# Answering Clinical Questions Using Machine Learning: Should We Look at Diastolic Blood Pressure When Tailoring Blood Pressure Control?

**DOI:** 10.3390/jcm11247454

**Published:** 2022-12-15

**Authors:** Maciej Siński, Petr Berka, Jacek Lewandowski, Piotr Sobieraj, Kacper Piechocki, Bartłomiej Paleczny, Agnieszka Siennicka

**Affiliations:** 1Department of Internal Medicine, Hypertension and Vascular Diseases, Medical University of Warsaw, Banacha 1a, 02-097 Warsaw, Poland; 2Department of Information and Knowledge Engineering, Faculty of Informatics and Statistics, Prague University of Economics and Business, W. Churchill Sq. 4, 120 00 Prague, Czech Republic; 3Department of Physiology and Pathophysiology, Wroclaw Medical University, Chałubińskiego 10, 50-368 Wroclaw, Poland

**Keywords:** cardiovascular risk, diastolic blood pressure, machine learning, SPRINT trial

## Abstract

**Background**: The guidelines recommend intensive blood pressure control. Randomized trials have focused on the relevance of the systolic blood pressure (SBP) lowering, leaving the safety of the diastolic blood pressure (DBP) reduction unresolved. There are data available which show that low DBP should not stop clinicians from achieving SBP targets; however, registries and analyses of randomized trials present conflicting results. The purpose of the study was to apply machine learning (ML) algorithms to determine, whether DBP is an important risk factor to predict stroke, heart failure (HF), myocardial infarction (MI), and primary outcome in the SPRINT trial database. **Methods**: ML experiments were performed using decision tree, random forest, k-nearest neighbor, naive Bayesian, multi-layer perceptron, and logistic regression algorithms, including and excluding DBP as the risk factor in an unselected and selected (DBP < 70 mmHg) study population. **Results**: Including DBP as the risk factor did not change the performance of the machine learning models evaluated using accuracy, AUC, mean, and weighted F-measure, and was not required to make proper predictions of stroke, MI, HF, and primary outcome. **Conclusions**: Analyses of the SPRINT trial data using ML algorithms imply that DBP should not be treated as an independent risk factor when intensifying blood pressure control.

## 1. Introduction

Diastolic blood pressure (DBP) is often a neglected parameter when making decisions concerning the optimization of blood pressure (BP) control aimed at reducing the risk of cardiovascular (CV) events. Most of the trials and recommendations focus on systolic blood pressure (SBP), leaving DBP values as a subject for discussion [1,2]. However, discussion concerning the role of DBP and the DBP-related J-curve is still relevant. Previously, it was shown that lower DBP may be related to the increased rate of myocardial infarction in hypertensive subjects [3]. The results of other studies addressing this issue confirm that intensive DBP reduction may increase the risk of coronary artery disease [4]. As demonstrated by the results of interventional clinical trials, low DBP may be related to increased CV risk; however, it cannot be interpreted as an independent CV risk factor [5,6]. Contrary to the results of clinical trials obtained using multiple statistical approaches, data from registries [7,8] and, importantly, common clinical concerns seem to point in another direction. Furthermore, the concern that lower DBP may be harmful can promote treatment inertia, resulting in poor BP control at an individual and population level.

As the problem of DBP relevance continues to remain a hot topic in hypertension, an alternative methodological approach incorporating artificial intelligence methods was used to strengthen the data that low DBP should not be the reason to abandon low SBP targets during treatment of hypertension. Machine learning (ML) methods focus on the prediction of an outcome based on variables, while statistical methods focus on the inference between variables and outcomes. Therefore, machine learning methods have different goals than conventional statistics, and might be more suitable to answer the question posed above.

The purpose of the current study was to apply machine learning algorithms to determine whether DBP is an important risk factor to predict stroke, heart failure (HF), myocardial infarction (MI), and primary cardiovascular outcome in the Systolic Blood Pressure Intervention Trial (SPRINT) database.

## 2. Methods

The SPRINT was a randomized, multi-center open-label trial aimed to verify the hypothesis that a lower SBP target would reduce clinical events [9]. The SPRINT included 9361 patients aged 50 and older with a screening SBP of 130 to 180 mmHg. Patients were included in the trial if they had at least one of the following: an increased CV disease (CVD) risk, defined as a history of clinical or subclinical CVD; chronic kidney disease (estimated glomerular filtration rate 20–60 mL min^−1^ 1.73 m^−2^); 10-year Framingham CVD risk of 15% or higher; or if they were 75 or older. Patients were excluded from the SPRINT if they had diabetes mellitus, heart failure, or a previous stroke. The SPRINT showed that achieving SBP of less than 120 mmHg, rather than 140 mmHg, contributed to a reduced risk of fatal and nonfatal major cardiovascular events and overall mortality. Primary outcome, myocardial infarction, acute coronary syndrome, stroke, congestive heart failure, and cardiovascular death were significantly reduced in the intensive BP (SBP < 120 mmHg) management arm compared with the standard management arm (SBP < 150 mmHg) (hazard ratio 0.75, 95% CI 0.64–0.89; *p* < 0.0001). The rationale, protocol, and results of the trial were published and widely discussed elsewhere [9,10].

We used the SPRINT trial data obtained from the NHLBI Biologic Specimen and Data Repository Information Coordinating Center (accession number HLB02021921a). The data are available upon reasonable request. The authors of this manuscript have no right to share the data. This manuscript does not necessarily reflect the opinions or views of the SPRINT Research Group or the NHLBI. The analysis was approved by the Ethics Committee at the Medical University of Warsaw.

The study rationale was based on the assumption that adding DBP to a ML model will not change model performance in classifying whether the outcome (stroke, MI, HF, primary outcome) is present or absent. Consequently, it would imply that the role of DBP is not relevant in predicting the events. Machine learning (ML) experiments were performed using the RapidMiner data science software platform [11]. The variables—age, sex, history of clinical cardiovascular disease, chronic kidney disease, allocation to the treatment arm, smoking status, heart rate, in-trial SBP, and in-trial DBP—which were used previously in a multivariate Cox model, were chosen out of the SPRINT database [5,6]. All experiments were performed separately for target (outcome) variables: stroke, myocardial infarction (MI), heart failure exacerbation (HF), and primary composite outcome (primary). The primary composite outcome was defined as the occurrence of a myocardial infarction, acute coronary syndrome other than myocardial infarction, stroke, acute decompensated heart failure, or death from a cardiovascular cause.

Individual phases of the ML experiments were performed as follows (Figure 1). Firstly, data were retrieved from the database. Then, the role of all variables was specified. As the values of target variables were imbalanced, the data was up-sampled using the synthetic minority oversampling technique (SMOTE) algorithm. The SMOTE algorithm increases the number of minority-class examples by artificially creating new examples similar to those existing in the original data [12]. In the up-sampled dataset, cases with DBP < 70 mmHg were selected. All further steps were performed for unselected (i.e., all cases) and selected (i.e., cases with DBP < 70 mmHg) populations. Each of the experiments was run separately for each outcome—primary, stroke, MI, and HF. Tenfold cross-validation was used to train and test the classification models. The training consisted of running a particular ML algorithm. We used decision tree (with gain ratio as the splitting criterion, maximum depth 10 and pruning), random forest (100 trees with gain ratio as the splitting criterion, maximum depth 10 and pruning), k-nearest neighbor classifier (with k = 5, weighted voting and mixed Euclidean distance), naive Bayesian classifier, multi-layer perceptron (with single hidden layer), and logistic regression. The performance of each classifier was evaluated using overall accuracy (denoted as accuracy in Table 1 and Table 2), the AUC for the detection of positive outcomes (denoted as AUC in Table 1 and Table 2), the mean of the F-measure for positive and negative outcomes (denoted as mF1 in Table 1 and Table 2), and the weighted F-measure for positive and negative outcomes (denoted as wF1 in Table 1 and Table 2). These are standard measures used to evaluate classification models [13,14]. The accuracy, mF1, and wF1 measures were used to evaluate the model as a whole, while the AUC was used to evaluate the performance of the model when predicting the presence of a corresponding diagnosis. Each of the experiments was performed with and without the DBP variable as the predictor.

### Statistical Analysis

This is a retrospective analysis of the SPRINT data. The performance of each machine learning algorithm for each of the outcomes (primary, stroke, MI, and HF), expressed by accuracy, AUC, mF1, and wF1 measures, was compared when taking into account the presence or absence of the DBP variable as the predictor. The paired two-sided *t*-test was used to compare the measures of the classifier performance with and without DBP as the predictor. The Welch *t*-test was also used for the comparison of classifiers’ performance in the selected and unselected population. The differences were considered significant at *p* < 0.05. All statistical computations were performed in an R 4.1.0 environment [15]. Standard ‘ggplot2’ and ‘tidyvers’ packages were used [16,17].

## 3. Results

The baseline characteristic of the study population was presented in detail in the original SPRINT report [9]. Appendix A present patients’ baseline characteristics and results in the intensive and standard treatment arms. The performance of each of the classifiers in the detection of analyzed outcomes is presented in Table 1 and Table 2. In the analysis of the unselected population, there were no significant differences in the performance of all of the classifiers when DBP was included as the predictor (the same was true for the opposite case with all outcome variables examined) (Figure 2). In the subpopulation of patients with DBP < 70 mmHg, the same relation occurred as mentioned above (Figure 3). There was also no difference detected in the performance of the classifiers when the selected and unselected populations were compared for all outcome variables, and when DBP was included and excluded as the variable (Figure 4). There were however significant differences when evaluating the particular machine learning algorithms. Regardless of the target class and experimental setting (excluding or including DBP, when using the whole population or the selected population with DBP < 70 mmHg), the multi-layer perceptron was always the best model and DT was the least effective one (Figure 5).

The results of the ML experiments performed show that variables were properly selected for the analysis with good goals of the prediction measures (Table 1 and Table 2). Models selected for the experiments showed consistent results of the performance measures (Table 1 and Table 2).

## 4. Discussion

The main finding of the study is that adding DBP as a factor does not affect the classifier performance in the prediction of stroke, MI, HF, and primary CV outcomes in the high-CV-risk populations of the SPRINT. DBP as a factor also does not improve the performance of the ML models. Even in the subpopulation of patients with DBP < 70 mmHg, adding DBP to the ML models did not improve the classifier performance. The foregoing is indirect evidence that DBP is not an important risk factor to predict CV outcomes. The question remains whether such a generalization might be made. Observational studies show the J-curve relation of DBP and cardiovascular events, especially in respect of the population with coronary artery disease [18,19,20]. A recent analysis of the EPHESUS randomized study in patients with coronary artery disease and heart failure showed an increased risk for all-cause death, cardiovascular death, and cardiovascular hospitalization in the subpopulation of subjects without reperfusion and with DBP lower than 70 mmHg [21]. The SPRINT secondary analysis also demonstrated there was a J-shaped association between the follow-up DBP and the composite cardiovascular outcome, regardless of intensive or standard treatment [22]. Additionally, although the authors of the study do no exclude reverse causality with respect to low DBP, such results are in line with the intuition of the practitioner associating reduced perfusion of organs with clinical outcomes. The results of analysis of other randomized studies do not confirm that rationale [23,24,25]. Recent studies using the mendelian randomization method [26,27] showed a linear, but not J-curved, relation between MI and DBP, and did not prove causality between DBP and CV events. The results of the current study reinforce those findings. In the currently published analysis of the SPS3 trial performed by Shihab et al. [28], the authors found that even in patients with low baseline DBP and previous stroke, intensive SBP lowering did not increase the risk of stroke. The commented paper has been acknowledged in recent research showing adequate perfusion of the brain in MRI in patients with low DBP values [29]. Our findings support those of Shihab et al. [28], as DBP in the chosen ML models did not improve stroke predictions.

A different methodological approach is applied in machine learning analysis in contrast to statistical analysis. ML prediction results are now widely used in consumer and medical decision-making algorithms, showing high efficiency. In the field of cardiology and imaging medicine, practitioners use ML algorithms [30,31,32,33,34]; yet, clinical study data are not often analyzed in that way [35,36]. It has been changing, though. Currently, new research results based on analysis of the SPRINT and ACCORD trial data incorporating ML methods demonstrated potential for the individualization of hypertension treatment [37].

Analysis performed previously by our team concerning the relationship between low DBP and the risk of cardiovascular events using a classic statistical approach [5,6,38] provided evidence supporting the findings of other researches. However, the current analysis might more effectively finally answer the question posed in the introduction. We believe that ML methods may add value or change researchers’ perspectives in cases where contrary results appear in various classical statistical methods. Nevertheless, we are far from treating ML methods as giving definite answers. As ML and AI methods are now increasingly used to develop new ways of diagnosing and evaluating clinical results, a critical approach to this methodology is necessary. Recommendations summarizing how to conduct and report the course of an experiment involving these methods have recently been published [39]. Since the authors of the present manuscript agree with this new approach, answers to the questions suggested in the recommendations are presented below in Table 3.

In the current study, experiments were performed using the chosen ML classifiers of various types. Other ML methods such as support vector machine (SVM) methods did not perform better than those chosen, but the future use of alternative classifiers such as the (artificial neural network (ANN) or the XBoost regressor does not exclude obtaining different results. However, it is worth mentioning that the study did not focus on finding the best classifier to detect CV outcomes. The objective of the experiments was to explore the role of DBP as a variable, and it was found that it is not a significant factor. Interestingly, adding DBP as a factor did not change the effect of predictions using both the best- and the worst-performing classifiers.

The current study has some limitations. Firstly, it was based on a single randomized study. Although it included 9361 patients, the study population did not cover the full spectrum of CVD risks, as patients with diabetes were excluded. Research using more complex databases and different variables is needed to fully establish the role of DBP as a CVD risk factor. As mentioned in the Methods section, variables selected for the current experiments were the same as those used previously in the multivariate Cox model [5,6]. If different variables were used, we could not exclude the possibility of a positive impact of DBP on outcomes (this was not the case in our study, though). In that case, the positive impact would be an effect of the different variable selection and not an effect of the ML method used. Current experiments were also performed with the additional risk factor of increased pulse pressure. No differences were noted in the study results after the inclusion of this parameter, and therefore the data are not presented in the tables and figures.

In the original database, a significant imbalance in the count of positive and negative outcomes was noted. Therefore, automatically performed up-sampling (SMOTE) was applied. Consequently, ML experiments were performed on the up-sampled SPRINT database which—even though the SMOTE procedure is designed for such purposes—may affect study results.

In conclusion, the study shows that results of the randomized clinical trials may be analyzed using ML methods and such analysis may have clinical implications. Our analysis using various types of ML models shows that adding DBP to the model does not improve model performance. This may imply that DBP should not be treated as an independent risk factor in predicting CV outcomes. The methodological approach which was used in the current study may be implemented in other cases.

## Figures and Tables

**Figure 1 jcm-11-07454-f001:**
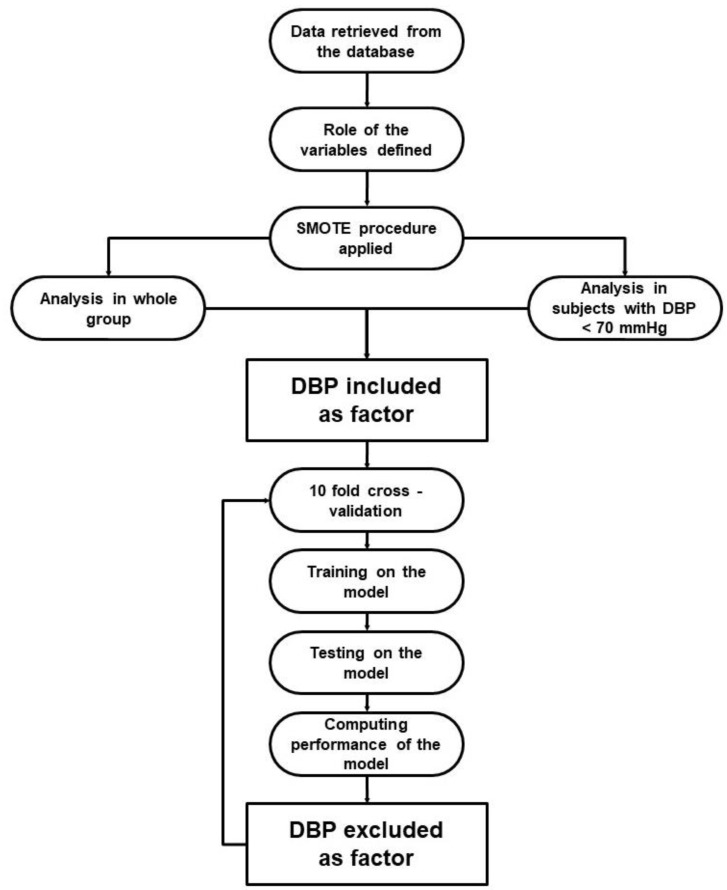
Flowchart presenting ML experiment phases. DBP—diastolic blood pressure, SMOTE—synthetic minority oversampling technique.

**Figure 2 jcm-11-07454-f002:**
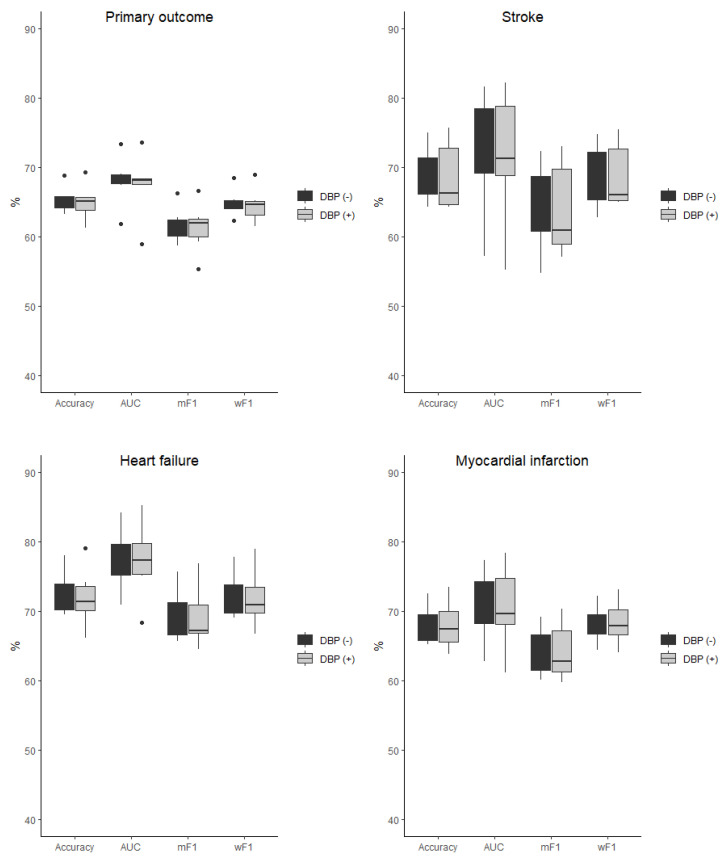
Comparison of the performance of the classifiers when DBP was included (DBP (+)) and excluded (DBP (−)) as the predictor presented as boxplots for all outcome variables examined in the unselected population. AUC—area under curve, mF1—mean F-measure, wF1—mean weighted F-measure, no differences detected for all of the comparisons.

**Figure 3 jcm-11-07454-f003:**
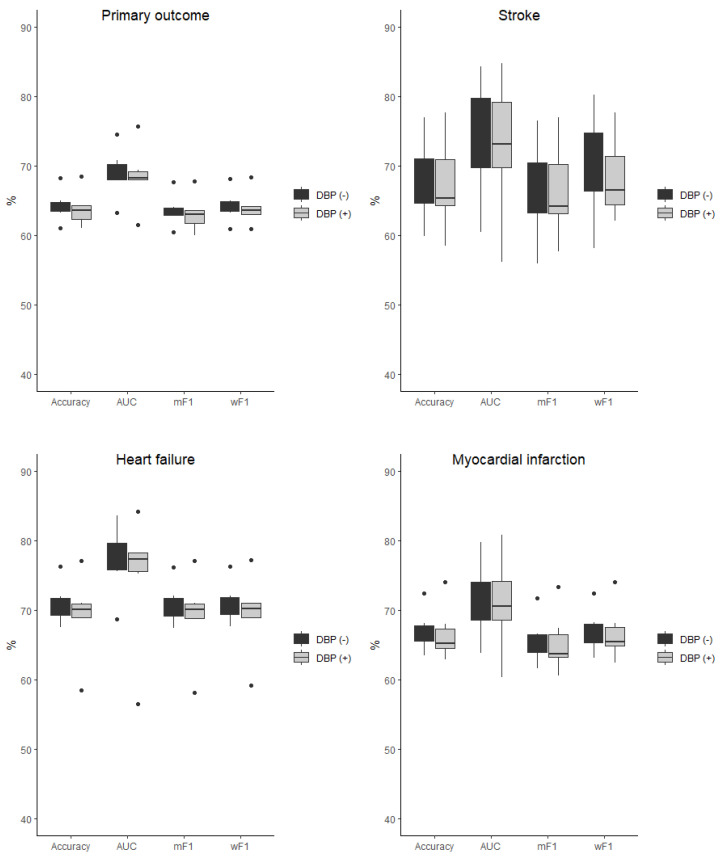
Comparison of the performance of the classifiers when DBP was included (DBP (+)) and excluded (DBP (−)) as the predictor presented as boxplots for all outcome variables examined in the selected population of subjects with DBP < 70 mmHg. AUC—area under curve, mF1—mean F-measure, wF1—mean weighted F-measure, no differences detected for all of the comparisons.

**Figure 4 jcm-11-07454-f004:**
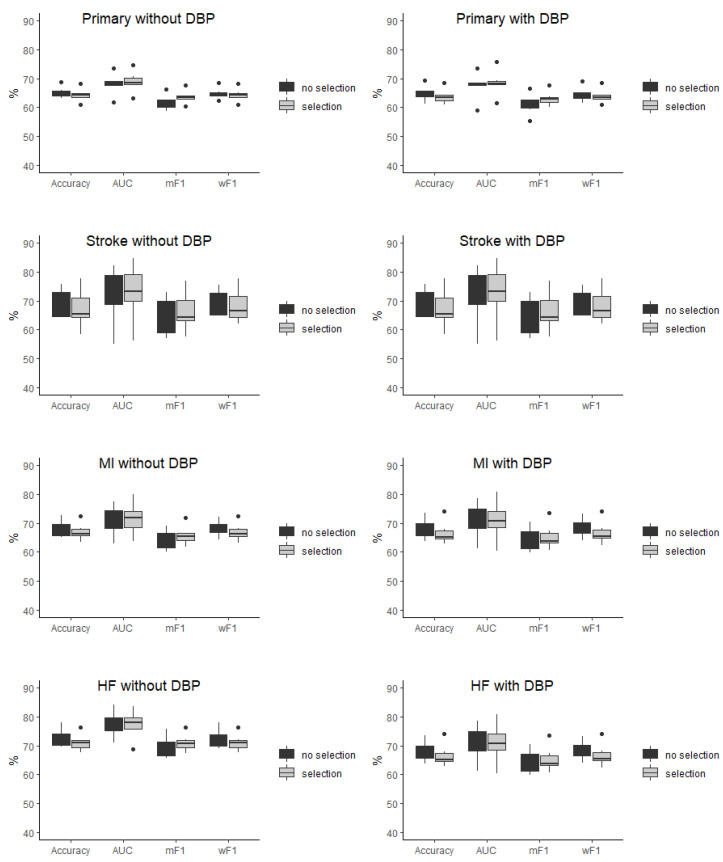
Comparison of the performance of the classifiers in the selected (selection) and unselected (no selection) population presented as boxplots for all of outcome variables when diastolic blood pressure was included and excluded as the predictor. AUC—area under curve, mF1—mean F-measure, wF1—mean weighted F-measure, no differences detected for all of the comparisons.

**Figure 5 jcm-11-07454-f005:**
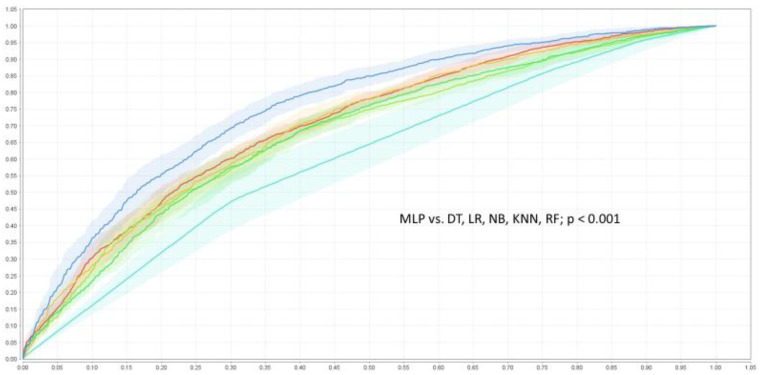
ROC curves of the classifiers detecting of primary outcome when DBP was included as the risk factor. DT—decision tree 

, KNN—k-nearest neighbor classifier 

, LR—logistic regression 

, MLP—multi-layer perceptron 

, NB—naive Bayesian classifier 

, RF—random forest 

.

**Table 1 jcm-11-07454-t001:** Performance of the classifiers for different outcomes, excluding and including diastolic blood pressure in the model for the whole population.

	Model Excluding DBP				Model Including DBP			
ML Algorithm	Accuracy (%)	AUC (%)	mF1 (%)	wF1 (%)	Accuracy (%)	AUC (%)	mF1 (%)	wF1 (%)
PRIMARY OUTCOME								
DT	63.2	61.9	58.7	62.3	63.6	59.0	59.3	62.7
RF	64.1	69.1	59.5	64.3	61.3	68.1	55.4	61.5
KNN	64.4	67.4	61.8	64.1	64.5	67.4	62.1	64.3
NB	65.9	68.6	62.7	65.4	65.6	68.3	62.8	65.2
MLP	68.9	73.4	66.3	68.5	69.3	73.6	66.6	68.9
LR	65.6	68.3	61.7	64.9	65.7	68.4	61.8	65.1
STROKE								
DT	64.3	57.2	54.7	62.8	64.3	55.2	57.1	65.0
RF	66.9	76.3	63.2	70.0	64.3	72.6	58.2	66.1
KNN	73.0	79.2	70.6	73.0	74.8	81.0	72.8	74.9
NB	66.0	68.9	61.0	65.2	65.7	68.5	60.9	65.0
MLP	75.0	81.7	72.3	74.8	75.7	82.2	73.1	75.5
LR	66.8	70.0	60.7	65.7	66.9	70.0	60.9	65.9
MYOCARDIAL INFARCTION								
DT	65.3	62.8	60.1	64.4	65.0	61.1	59.8	64.1
RF	65.2	72.1	61.4	68.3	63.9	70.8	61.2	68.7
KNN	70.0	75.1	67.4	69.9	70.8	76.1	68.3	70.7
NB	68.3	68.9	64.2	67.7	67.6	68.5	63.8	67.1
MLP	72.6	77.4	69.2	72.2	73.6	78.4	70.4	73.2
LR	67.3	68.0	61.8	66.4	67.3	68.0	61.8	66.4
HEART FAILURE								
DT	69.5	71.0	65.7	69.0	66.2	68.4	64.5	66.8
RF	73.2	79.5	69.5	72.9	71.8	78.2	67.5	71.5
KNN	74.2	79.7	71.9	74.2	74.2	80.4	72.0	74.2
NB	70.1	75.0	66.6	69.7	69.8	75.1	66.8	69.6
MLP	78.0	84.2	75.7	77.9	79.1	85.3	76.9	78.9
LR	70.7	75.9	66.6	70.2	71.0	76.4	67.0	70.4

AUC—area under curve, DT—decision tree, KNN—k-nearest neighbor classifier, LR—logistic regression, mF1—mean F—measure, MLP—multi-layer perceptron, NB—naive Bayesian classifier, RF—random forest, wF1—mean weighted F-measure.

**Table 2 jcm-11-07454-t002:** Performance of the classifiers for different outcomes, excluding and including diastolic blood pressure in the model for the population with DBP < 70 mmHg.

	Model Excluding DBP				Model Including DBP			
ML Algorithm	Accuracy (%)	AUC (%)	mF1 (%)	wF1 (%)	Accuracy (%)	AUC (%)	mF1 (%)	wF1 (%)
PRIMARY OUTCOME								
DT	61.1	63.2	60.4	61.0	61.1	61.5	60.0	60.9
RF	65.0	70.8	64.1	65.0	62.0	69.4	61.4	63.0
KNN	63.3	68.6	62.8	63.3	63.0	68.2	62.6	63.0
NB	64.2	68.4	63.7	64.2	64.2	68.4	63.6	64.1
MLP	68.2	74.5	67.7	68.2	68.5	75.7	67.8	68.4
LR	64.3	67.9	63.4	64.2	64.3	68.0	63.4	64.2
STROKE								
DT	59.9	60.5	56.0	58.1	58.5	56.1	57.7	62.1
RF	68.1	78.9	67.2	72.6	64.7	76.5	64.3	67.3
KNN	72.1	80.1	71.5	75.5	72.6	80.1	72.2	72.8
NB	64.3	69.8	63.0	66.2	64.2	69.8	62.8	64.0
MLP	77.0	84.3	76.5	80.2	77.7	84.8	77.0	77.7
LR	65.7	69.9	63.8	66.7	66.0	69.9	64.0	65.7
MYOCARDIAL INFARCTION								
DT	63.5	63.8	61.7	63.2	62.8	60.3	60.6	62.5
RF	68.1	73.9	66.6	68.3	64.4	72.0	63.2	65.8
KNN	66.9	74.1	66.3	67.0	68.0	74.9	67.5	68.2
NB	65.8	69.6	64.6	65.7	65.0	69.3	63.7	64.8
MLP	72.4	79.8	71.7	72.4	74.1	80.9	73.4	74.1
LR	65.5	68.3	63.8	65.2	65.5	68.4	63.7	65.2
HEART FAILURE								
DT	67.6	68.7	67.5	67.6	58.5	56.5	58.1	59.1
RF	70.8	79.9	70.8	70.9	69.4	78.3	69.3	69.4
KNN	71.9	79.2	72.1	72.1	70.9	78.2	71.0	71.1
NB	68.8	75.6	68.7	68.9	68.8	75.2	68.7	68.8
MLP	76.2	83.6	76.2	76.3	77.2	84.2	77.2	77.2
LR	71.0	76.6	70.8	71.0	71.0	76.6	70.9	71.0

AUC—area under curve, DT—decision tree, KNN—k-nearest neighbor classifier, LR—logistic regression, mF1—mean F—measure, MLP—multi-layer perceptron, NB—naive Bayesian classifier, RF—random forest, wF1—mean weighted F-measure.

**Table 3 jcm-11-07454-t003:** Questions about artificial intelligence-based prediction of cardiovascular disease.

**Is AI Needed to Solve the Targeted Medical Problem?** *In the Current Study ML Methods Were Used to Solve the Problem of the DBP Relevance as an Independent Risk Factor. Previous Analysis Using Classical Methods Gave Ambiguous Results.*
How does the AI prediction model fit in the existing clinical workflow? *The current analysis is a secondary analysis of the clinical trial. It does not apply to any specific existing clinical workflow.*
Are the data for prediction model development and testing representative for the targeted patient population and intended use? *The SPRINT population was representative for the population of patients with high cardiovascular risk requiring antihypertensive treatment.*
Is the (time)point of prediction clear and aligned with the feature measurements? *The current analysis is a secondary analysis of the clinical trial.*
Is the outcome variable labelling procedure reliable, replicable and independent? *It is secondary analysis of the clinical trial. We used labeling of the original SPRINT database.*
Was the sample size sufficient for AI prediction model development and testing? *The sample size of the analysis was determined by the sample size of the SPRINT. The environment used for the analysis did not reveal sample size issues. A posteriori sample size calculation was not performed.*
Is optimism of predictive performance of the AI prediction model avoided? *The performance of AI prediction models was tested.*
Was the AI model performance evaluated beyond simple classification statistics? *The created models were evaluated using 10-fold cross-validation. Various quality criteria was computed. Overall accuracy, mF1, and wF1 measures were used to evaluate the model as a whole, AUC was used to evaluate the model when predicting the diagnosis.*
Were the relevant reporting guidelines for AI prediction model studies followed? *The topics included in the TRIPOD statement were addressed in the manuscript.*
Is algorithmic (un)fairness considered and appropriately addressed? *SPRINT focuses on the population of patients with high cardiovascular risk requiring antihypertensive treatment and the results were biased towards this group of patients.*
Is the developed AI prediction model open for use, further testing, critical appraisal and updating, and use in daily practice? *Standard data mining platform RapidMiner was used so it is easy to perform retest.*
Are the presented relations between individual features and the outcome not overinterpreted? *Conclusions were drawn with caution.*

## Data Availability

Raw data are available upon reasonable request from NHLBI.

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
