# Peer review of "Answering Clinical Questions Using Machine Learning: Should We Look at Diastolic Blood Pressure When Tailoring Blood Pressure Control?"

_jcm, 2022, doi:10.3390/jcm11247454_

Round 1

Reviewer 1 Report

The manuscript as a whole needs to be rechecked for grammar, flow, and syntax. In addition, more than one individual wrote the manuscript due to inconsistencies in writing style. 

The abstract lacks clarity (only one sentence for the background) and needs to be expounded upon or restructured.

Additional background information needs to be added to the introduction.

Figures 2-4 do not need the p-values in the figure—a significant symbol (*) above the significant markers needs to be inserted. 

Exclude having tables and figures referenced in the first paragraph of the discussion. Instead, present your main findings and then expound upon the finds.

Based on the substantial results, the discussion does not fully address the results against the current literature. Further detailed discussion is needed. This may be the shortest discussion for a manuscript I have ever seen. 

Reviewer 2 Report

In this article authors have used various ML models to predict the implication of DBP on cardiovascular outcomes. This work is relevant and I have the following recommendations

1. Authors need to adhere to prior guidelines for reporting and describing ML methodology " Refer Maarten van Smeden, Georg Heinze, Ben Van Calster, Folkert W Asselbergs, Panos E Vardas, Nico Bruining, Peter de Jaegere, Jason H Moore, Spiros Denaxas, Anne Laure Boulesteix, Karel G M Moons, Critical appraisal of artificial intelligence-based prediction models for cardiovascular diseaseEuropean Heart Journal, Volume 43, Issue 31, 14 August 2022, Pages 2921–2930"

2. Authors need to discuss how the different ML models performed with predicting accuracy, compairison among the model. Which was worst, which was better?

3. Did the authors try artificial neural network?

4. Depict the image of AUC of the models, the diagram of forest tree for the primary outcome in image

5. I think the conclusion is not correct, authors can only state that multiple ML based algorithms were not able to establish the relevance of DBP in primary outcome as studied by the authors based on the SPRINT data base. Future studies with better models (ANN), more variables, larger sample size might be able to identify implications of DBP on MACE
